# Assessment of Geographical Distribution of Emerging Zoonotic *Toxoplasma gondii* Infection in Women Patients Using Geographical Information System (GIS) in Various Regions of Khyber Pakhtunkhwa (KP) Province, Pakistan

**DOI:** 10.3390/tropicalmed7120430

**Published:** 2022-12-11

**Authors:** Muhammad Jamil Khan, Murad A. Mubaraki, Sarwat Jahan, Baharullah Khattak, Manzoor Khan, Malik Abid Hussain Khokhar, Ijaz Ahmad

**Affiliations:** 1Department of Animal Sciences, Facility of Biological Sciences, Quaid-I-Azam University, Islamabad 45320, Pakistan; 2Clinical Laboratory Sciences Department, College of Applied Medical Sciences, King Saud University, Riyadh 11362, Saudi Arabia; 3Department of Microbiology, Kohat University of Science and Technology, Kohat 26000, Pakistan; 4Department of Statistics, Quaid-I-Azam University, Islamabad 45320, Pakistan; 5Department of Biochemistry, KMU Institute of Medical Sciences, Kohat 26000, Pakistan; 6Department of MAS (NIPCONS), National University of Sciences & Technology (NUST), Islamabad 44000, Pakistan; 7Department of Chemistry, Kohat University of Science and Technology, Kohat 26000, Pakistan

**Keywords:** zoonotic infection, diagnosis, regional variations, divisions and districts

## Abstract

Toxoplasmosis is a zoonotic parasitic disease caused by *T. gondii*, an obligate intracellular apcomplexan zoonotic parasite that is geographically worldwide in distribution. The parasite infects humans and all warm-blooded animals and is highly prevalent in various geographical regions of the world, including Pakistan. The current study addressee prevalence of Toxoplasma infection in women in various geographical regions, mapping of endemic division and t district of Khyber Pakhtunkhwa province through geographical information system (GIS) in order to locate endemic regions, monitor seasonal and annual increase in prevalence of infection in women patients. Setting: Tertiary hospitals and basic health care centers located in 7 divisions and 24 districts of Khyber Pakhtunkhwa (KP) province of Pakistan. During the current study, 3586 women patients from 7 divisions and 24 districts were clinically examined and screened for prevalence of *T. gondii* infection. Participants were screened for Toxoplasma infection using ICT and latex agglutination test (LAT) as initial screening assay, while iELISA (IgM, IgG) was used as confirmatory assay. Mapping of the studied region was developed by using ArcGIS 10.5. Spatial analyst tools were applied by using Kriging/Co-kriging techniques, followed by IDW (Inverse Distance Weight) techniques. Overall prevalence of *T. gondii* infection was found in 881 (24.56%) patients. A significant (<0.05) variation was found in prevalence of infection in different divisions and districts of the province. Prevalence of infection was significantly (<0.05) high 129 (30.07%) in Kohat Division, followed by 177 (29.06%), 80 (27.87%), 287 (26.72%), 81 (21.21%), 47 (21.07%), and 80 (13.71%) cases in Hazara Division, D.I Khan Division, Malakand Division, Mardan Division, Bannu Division, and Peshawar Division. Among various districts, a significant variation (<0.05) was found in prevalence of infection. Prevalence of infection was significantly (<0.05) high 49 (44.95%) in district Karak, while low (16 (10.81%) in district Nowshera. No significant (>0.05) seasonal and annual variation was found in prevalence of Toxoplasma infection. LAT, ICT and ELISA assays were evaluated for prevalence of infection, which significantly (<0.05) detected *T. gondii* antibodies. LAT, ICT and ELISA assays significantly (<0.05) detected infection, while no significant (>0.05) difference was found between positivity of LAT and ICT assays. A significant difference (<0.05) was found in positivity of Toxoplasma-specific (IgM), (IgG) and (IgM, IgG) immunoglobulin by ICT and ELISA assay. The current study provides comprehensive information about geographical distribution, seasonal and annual variation of Toxoplasmosis infection in various regions of Khyber Pakhtunkhwa province of Pakistan. Infection of *T. gondii* in women shows an alarming situation of disease transmission from infected animals in the studied region, which is not only a serious and potential threat for adverse pregnancy outcomes, but also cause socioeconomic burden and challenges for various public and animal health organizations in Pakistan and across the country.

## 1. Introduction

Many emerging human diseases are zoonotic. Toxoplasma infection is one of zoonotic parasitic infection caused by a protozoan parasite *T. gondii* [1]. *Toxoplasma gondii* infects humans and various other animal species. The parasite causes a common zoonotic parasitic infection (Toxoplasmosis) in humans and is geographically worldwide in distribution [2]. The parasite is capable of infecting all warm-blooded animals and is highly prevalent in various geographical regions of the world [3]. The parasite completes its life cycle in two different hosts, i.e., sexual part occurs in domestic and wild felids, while asexual part occurs in any mammal host [4]. The parasite is intracellular in nature and globally has a medical and veterinary importance. Human is mainly infected through ingesting undercooked or raw animal meat that contains viable tissue cysts, or ingesting oocysts in various contaminated foods, animal milk, or water [5]. The route of parasite transmission may vary according to human habits in various geographical regions of the world [6]. Transmission of parasite may occur from contaminated utensils, knives, or raw meat. Drinking water or animal milk contaminated with cysts, organ transplantation, transfusion of blood and nosocomial (needle-stick injury) have also been reported as a source of parasite transmission. Human infection can occur through direct ingestion of parasitic oocysts, during hand-to-mouth contact after cleaning cat’s litter box, gardening, or contact with children’s sandpits and touching anything contaminated with fecal material of cats. Congenital infection may occur from infected mother to her newborn child [7]. Infection is usually asymptomatic or has mild flu-like clinical symptoms, but it may cause serious adverse obstetric outcomes, such as abortion, stillbirth, premature childbirth, or serious fetal damage during pregnancy [8]. Infection is routinely diagnosed through various serological agglutination tests [9], while PCR, histopathology, IFAT, ELISA, DAT and MAT are usually used for confirmation of selected clinical cases, prevalence and surveillance of infection in animals and humans [10]. ELISA assay is currently used for proper detection of *T. gondii* specific IgM and IgG antibodies in sera of infected individuals [11]. Recent patterns of human-driven change in environment and globalization of trade and travels have enhanced spillover and spillback of *T. gondii* parasites into human population, facilitating its further propagation to local, regional and global communities of human [12,13]). In Pakistan, large number of human population has low socioeconomic status, due to which women are usually more exposed to different infections [14], including infection caused by *T. gondii*. Despite important zoonotic infection, no comprehensive data is properly available on *T. gondii* infection in women patients in Khyber Pakhtunkhwa province of Pakistan. The current study was designed to determine actual prevalence of *T. gondii* infection in women patients and collect information regarding geographical distribution seasonal and annual variation in prevalence of infection in women patients using GIS and remote sensing techniques in various regions of Khyber Pakhtunkhwa province of Pakistan, which will globally provide essential, comprehensive and updated information for development of public health polices for proper monitoring, early diagnosis managment and prevention of human infection.

## 2. Materials and Methods

**Study Area:** The province Khyber Pakhtunkhwa (KP) is situated in the northwestern region of Pakistan with Pak-Afghan border, which is geographically located at a latitude and longitude 34.0000° North, 71.3200° East. The total area of province is 74,521 km^2^ (28,773 sq mi), while total human population is 30,523,371. Khyber Pakhtunkhwa is connected at North-East to Gilgite Baltistan, North-West to Afghanistan, East to Azad Kashmir, South-East to Punjab, and South to Baluchistan provinces of Pakistan (Figure 1). Khyber Pakhtunkhwa province consists of 26 districts and 2989 village councils. The climate of province can be classified as tropical monsoon. Despite much of the northern part of the province lying beyond the tropical zone, the whole province has a tropical climate, which is marked by relatively high temperatures and dry winters. There are four seasons in the province, spring (March–April), summer (May–September), autumn (October), and winter (November–February) [15].

**Study design:** The study was conducted from January 2017 to March 2020 in 7 divisions and 24 districts located in Khyber Pakhtunkhwa (KP) province of Pakistan. During the current study, various hospitals, health care units, maternity centers, basic health unit (BHU) and private health care centers were properly visited for collection of information and blood samples from suspected women patients, who were visiting for clinical or obstetrical examination. All enrolled patients were asked for collection of information, along with blood sample in a sterile disposable gel tube for assessment of *T. gondii*-specific IgM and IgG antibodies during laboratory examination.

**Blood collection:** Blood samples were aseptically taken from all enrolled patients and dispensed into gel tubes for *T. gondii* specific serological assays (LAT, ICT, IgM, IgG ELISA). These blood tubes were labeled with specific identification codes and analyzed by LAT, ICT assays on daily basis and IgM and IgG ELISA on weekly basis for confirmation of t positivity of LAT and ICT assays.

**Inclusion criteria:** All women patients were included, who were in reproductive age group (17–45 years old), married, pregnant or non-pregnant, visited hospital for clinical or obstetrical problems and to whom, concerned physician had recommended *T. gondii*-specific diagnostic assays, such as LAT, ICT, or ELISA due to various reproductive or obstetric problems and who were willingly participated in the current study. 

**Exclusion criteria:** Those patients who were ≤17 and >45 years old, unmarried menopausal age, or who fulfilled inclusion criteria, but were not willingly participate in our study.


**Serological Assays for Diagnosis of Infection:**


Toxoplasma Latex Agglutination Tests (LAT): The test was performed according to protocol of manufacturer (Fortress Diagnostics Limited, Antrim, UK). The sera of all patients were 1/16 diluted in physiological saline. Then, 1 drop of sera was placed on a black circle of slide and 1 drop of latex reagent was added over the testing sera and mixed well with a micropipette on the slide. The presence or absence of agglutination was observed within 3 min. Both positive and negative controls were used for validation of each testing batch. Sodium azide was used in 0.1% for positive control, and in 0.95 g/L for negative control.


**Toxo IgG/IgM Rapid Test Cassette Method:**


(Lateral Flow Chromatographic Immunoassay)

Toxo IgG/IgM rapid test cassette was a lateral flow chromatographic immunoassay that is used for qualitative detection of *T. gondii* specific antibodies (IgG and IgM) in patient sera. The test rapidly diagnose Toxoplasma infection. The assay was performed according to method of Chinese pharmacopeia and China biological products procedures. About 10 µL sera was transferred to the “S” well of test cassette and 2 drops buffer (about 70 µL) was mixed with sera in the test cassette and was left at room temperature for 15 min until colored lines appeared in the test cassette. The result was interpreted within 15 min. The appearance of IgG test line was considered presence of Toxoplasma-specific IgG antibodies and IgM test line showed presence of IgM antibodies. The appearance of both IgM and IgG was also considered as a positive test, while the test was considered negative, when only one colored line appeared in control negative region (C) and no colored line was observed in IgM or IgG test line region.

**Toxoplasma specific IgM, IgG ELISA:** LAT and ICT positive sera of women patients were finally analyzed for antibodies confirmation by commercial ELISA kit of Vircell microbiologist, Spain [16]. in 96-wells ELISA plate. *T. gondii* antigen strain RH (ATCC 50174) was used. Four wells were used for control, two for cut off serum and one for negative and positive control sera. ELISA assay was used according to manufacturer instructions. ELISA plate was read after recommended protocol by a spectrophotometer at 450/620 nm.

**Result Interpretation:** Mean O.D was calculated for cut-off serum and an antibodies index was measured by following formula of ELISA kit.

**Antibody index:** (sample O.D/cut-off serum mean O.D.) × 10. An antibody index <9 was negative, 9–11 equivocal and >11 positive for Toxoplasma infection.

**Designing GIS Maps:** GIS maps were developed by using ArcGIS 10.5. Before working in GIS environment, the data was sorted and it was ensured that each polygon had same value, as enumerated by the time of data tabulation. Later, one Spatial analyst tool was applied using Kriging/Co-kriging techniques, followed by IDW (Inverse Distance Weight) technique. Variogram and Semivariogram properties were also incorporated in the form of ordinary methods with the spherical Semivariogram model, where minimum RMS (root mean square) error was ensured. For the radius of cell size, 6–12 points were allocated. Moreover, the near neighborhood algorithm was used for IDW to depict the raster output of map. Shape files of the study area were prepared by digitizing the map as published by Survey of Pakistan, where the WGS (World Geographic System) 1984 as Datum and World Geographic Coordinated System 1984 were used. Generic symbology was used to adjust the maps with suitable color scheme ramps.

**Ethical aspects:** The study and recommended questioner was approved by the ethical board of Quaid-E-Azam University Islamabad, Pakistan, through approval record number of BEC-FBS-QAU 2019-145.

**Data analysis:** The Data was stored in Microsoft Excel 2007 program and was statistically analyzed by R Studio Version 1.2.1335 (2009–2019) and SPSS IBA program.

## 3. Results

During the current study, 3586 women patients of reproductive age were clinically examined and screened for prevalence of emerging zoonotic *T. gondii* infection using immunochromatographic test (ICT) and latex agglutination test (LAT) as initial screening assay, while indirect enzymes-linked immunosorbant assay (iELISA) was used as confirmatory assay for detection of *T. gondii*-specific immunoglobulin (IgM) and IgG. The overall prevalence of infection was found in 24.56%, 881/3586 women (Table 1).

Various socio-demographic characteristics, such as women’s age, residential and obstetric statuses were evaluated for prevalence of Toxoplasma infection (Table 1). A significant difference (<0.05) was found in prevalence of infection in residential statuses, while no significant (>0.05) difference was found in prevalence of infection in women of various age and obstetric group. Prevalence of infection was high 26.62% (152/571) in women, who were 37–45 year old, residents of rural areas 26.07% (675/2589), and non-pregnant 25.21% (635/2518) as compared to 24.18% (729/3015), 20.66% (206/997), and 23.03% (246/1068) positive cases in women, who were 17–36 year old, residents of urban areas and pregnant (Table 1).

Prevalence of Toxoplasma infection was determined in various geographical regions (7 divisions, 24 districts) of Khyber Pakhtunkhwa (KP) province of Pakistan (Table 2 and Table 3). Significant (<0.05) variations were found in prevalence of infection in various divisions. Among patients from Kohat Division, 30.7% (129)/429) cases were found positive, followed by 29.06% (177/609), 27.87% (80/287), 26.72% (287/1074), 13.71% (80/584), 21.21% (81/239), and 47 21.07% (47/223) cases from Hazara Division, D.I Khan Division, Malakand Division, Peshawar Division, Mardan Division and Bannu Division, respectively.

Prevalence of infection was also determined in various districts of the province (Table 3). A significant (<0.05) difference was found in prevalence of infection in 24 districts of Khyber Pakhtunkhwa (KP) province of Pakistan (Table 3).Prevalence of infection was high 44.95% (49/109) in women, who were living in district Karak, followed by 44.76% (47/105), 40.36% (90/223), 36.69% (51/139), 34.04% (64/188), 32.72% (36/110), 30.58% (26/85), 28.49% (55/193), 27.51% (33/120), 27.05% (46/170), 26.04% (50/192), 23.43% (30/128), 21.75% (52/239), 20.56% (29/141), 18.02% (31/172), 17.98% (34/189), 17.16% (23/134), 17.14% (24/140), 17.14 (18/105), 16.15% (16/148), 15.88% (17/107), 13.59 (14/103), 12.14% (30/247) and 10.81% (16/148) positive cases in district f Chitral, Swat, Mansehra, Dera Ismail Khan, Battagram, Kohistan, Malakand, Bannu, Abbottabad, Kohat, Hangu, Mardan, Swabi, Upper Dir, Charsadda, Buner, Lower Dir, Haripur, Tank, Shangla, Lakki Marwat, Peshawar and Nowshera, respectively (Figure 2).

Prevalence of infection was determined in different months and seasons during three-year study (Table 4), but no significant (>0.05) difference was found in prevalence of infection. Prevalence of infection was high 33.19% (80/241) in the month of September, while it was low 12.15% (40/329) in the month of July (Table 4).

Prevalence of Toxoplasma infection was determined in various seasons during three-year study, but no significant difference (>0.05) was found in rate of infection (Table 5). Among positive cases, 25.31% (380/1501) cases were found from the month of May to September (summer), followed by 25.22% (316/1253) cases from November to February (winter), while 25.01% (58/232) and 21.16% (127/600) cases were detected in October (autumn) and from the month of March to April (spring).

Assessment for annual prevalence of Toxoplasma infection was carried out, but no significant (>0.05) annual variation was found in prevalence of infection during three-year study (Table 6). Prevalence of infection was high 24.83% (263/1059) in the year 2019, which was followed by 24.68% (291/1179) and 24.25% (327/1348) cases in the years 2017 and 2018, respectively (Table 6).

Various diagnostic assays, such as latex agglutination test (LAT), immunochromatographic test (ICT) and enzyme-linked immunosorbant assay (ELISA), were evaluated for prevalence of Toxoplasma infection, which significantly detected *T. gondii* antibodies (Table 7). LAT, ICT and ELISA assays significantly (<0.05) detected Toxoplasma infection, while no significant (>0.05) difference was found between positivity of LAT and ICT assays. LAT assay was significantly (<0.05) positive in 25.46% (913/3586) cases, and ICT as a whole was significantly (<0.05) positive in 25.51% (915/3586) cases, while ICT IgM was positive in 3.12% (112/3586) and ICT IgG in 19.35% (694/3586) cases. ICT (IgM, IgG) was simultaneously positive in 3.04% (109/3586) cases. LAT and ICT assays were simultaneously positive in 25.46% (913/3586) women. ELISA significantly (<0.05) confirmed Toxoplasma-specific IgM and IgG antibodies in 73.72% (881/1195) women, with positivity of IgM in 8.03% (96/1195) and IgG in 57.07% (682/1195), while IgM and IgG were collectively found positive in 8.62% (103/1195%) patients. A significant variation (<0.05) was found in positivity of Toxoplasma-specific (IgM), (IgG) and (IgM, IgG) immunoglobulin by ICT and ELISA assays, while a significant (<0.05) difference was also found in positivity among IgM, IgG and IgM, and IgG by ELISA assays (Table 7).

## 4. Discussion

The current study was the first study, which was conducted on geographical distribution of *T. gondii* infection in women patients, who visited to clinicians in hospitals and health care centers located in seven divisions and twenty-four districts in Khyber Pakhtunkhwa (KP) province of Pakistan (Table 3 and Figure 1), which highlighted comprehensive information related to overall prevalence of infection in women in various geographical regions of the province intra-division and intra-district variation in prevalence of infection (Table 3) and comparative assessment of various laboratory assays (Table 7) that are commonly used for diagnosis of *T. gondii* infection in Pakistan. The current results on prevalence of infection in women patients of various age, residential, and obstetric groups (Table 1) clearly indicated that *T. gondii* parasites persist in different pet and food animals in the studied regions of Pakistan and is silently infecting humans due to zoonotic nature of parasite, because infected animals are main reservoirs of parasite. Infection of *T. gondii* in different livestock and pet animal species has been previously reported by several investigators in Pakistan, like Ahmad and Tasawar, Perveen and Shah, Ahmad et al., Ahmed et al., Ahmad and Tasawar and Khan et al. [17,18,19,20,21,22], which clearly indicates that *T. gondii* parasite is transmitted from infected animals to women though different sources, because women of all age groups usually remain engaged with animal in homes for their care and management in the studied regions of Khyber Pakhtunkhwa province of Pakistan (Figure 1). They daily clean contaminated soil from animal shelters and surrounding areas of animals, which is a rich source of acquiring various zoonotic infections. Ajmal et al. [23] had previously detected *T. gondii* in different environmental matrices, such as soil, water, vegetable and fruits, which are considered risk factors for acquiring Toxoplasma infection. Due to poverty and lack of proper resources in the studied regions of Khyber Pakhtunkhwa province of Pakistan (Figure 1), most human families cannot afford filtered water and drink water from rivers, streams, or open water tanks that are contaminated with oocyst of *T. gondii* during dusting of homes or rainy seasons. In addition to dusting of homes and animal shelters, men, women, and children of all age groups usually consume roasted infected chicken from hotels and during marriage parties. These chickens are infected due to endemicity of infection in the studied province (Figure 1). Mahmood et al. [24] reported *T. gondii* in uncaged (20.70%) and caged (5.90%) chickens intended for human consumption from District Mardan. They stated that high prevalence in free-range chickens indicates environments and soil contaminated with oocysts of *T. gondii*, as free-range chickens are infected by feeding from the ground or soil contaminated with oocysts of *T. gondii*. People in the region prefer uncaged chicken, as they believe that uncaged chicken are tasty and full of vital forces, due to which they are usually infected. In addition to consumption of chicken, sources of drinking water, such as rivers, streams, pounds, rain, and open water tanks in the studied region, are usually contaminated due to environmental contamination. Ayaz et al. [25] and Khan et al. [26] detected parasites in different sources of drinking water in various districts of Khyber Pakhtunkhwa province of Pakistan. Anvari et al. [27] claimed that prevalence of *T. gondii* in cattle meat imported from Pakistan is higher as compared to indigenous livestock meat in South East of Iran. Persistence of these reported risk factors in the studied regions (Figure 1) strongly support our current prevalence 881 (24.56%) of *T. gondii* infection in women of Khyber Pakhtunkhwa province of Pakistan (Table 1). Infection has also been reported in women by many other investigators from selected regions of Pakistan. Faisal et al., Majid et al., Jan et al., Zeb et al., Aleem et al. and Sadiqui et al. [28,29,30,31,32,33] reported Toxoplasma infection from various limited regions of Khyber Pakhtunkhwa province of Pakistan, which supports our current data (Table 3). The current data on prevalence of infection in women in all districts of the province indicated that Pakistan is an endemic zone for zoonotic *T. gondii* infection, and risk factors also persist for transmission of parasite to humans. Jones et al. [34] and Xiao et al. [35] stated that prevalence of *T. gondii* infection in animals and human is used as an indicator for endemicity of pathogens in a particular geographical region in the world, which supports our claims. Majid et al. [29] reported various risk factors, such as close contact with cats and other livestock animals, consumption of unpasteurized milk, undercooked meat, raw eggs, vegetables, water sources, and residence in rural areas, which were significantly associated with human infection in various districts of Malakand division. 

During the current study, an overall prevalence of infection was recorded (24.56%) in women patients (Table 1), while significant (<0.05) variations were found in prevalence of infection in various division and districts of Khyber Pakhtunkhwa (KP) province of Pakistan (Table 2), which could be due to variations in various epidemiological variables, such as disease prevalence in food animals in different regions, method of determining and collecting sample sizes, socioeconomic conditions, levels of education, knowledge and practices regarding Toxoplasma infection, eating habits, consumption of raw meat, unwashed raw fruits, vegetables or farming as an occupation, general and reproductive health of patients, immunological condition, pregnancy, age and residential status (urban, rural) of patients. A significant (<0.05) difference was found in prevalence of *T. gondii* infection in women of various residential statuses. Prevalence of infection was high 675 (26.07%) in women, who lived in rural areas as compared to 206 (20.66%) positive cases in women, who lived in urban regions of the studied province (Table 1). In rural area of Khyber Pakhtunkhwa province of Pakistan, people mostly live in houses, which are made of local mud, where sanitation system is very poor and people are not educated and aware about various zoonotic diseases. Moreover, people depend on various species of livestock animals for their earning and they usually drink unpasteurized animal milk, which may contain tachyzoite stage of parasite. Mahmood et al. [24] stated that about 30–35 million rural population of human is engaged in livestock farming in Pakistan.

During the current study, prevalence of *T. gondii* infection was significantly (<0.05) different in various divisions and districts of Khyber Pakhtunkhwa province of Pakistan (Table 3), which is also supported by studies of different local investigators in published literature. Shah et al., Faisal et al., Majid et al., Jan et al., Zeb et al., Aleem et al., and Sadiqui et al.) [28,29,30,31,32,33,36] also reported *T. gondii* infection in humans with a prevalence of 28.44%, 19.25%, 65.71%, 21%, 2.5%, 47.2% and 24.8% cases from District Mardan, District Swabi, Malakand division, District Charsadda, District Peshawar, District Swat and Abbottabad and Mansehra district, while Shah et al. [36] reported 20% positive cases from other limited regions in Khyber Pakhtunkhwa (KPK) province of Pakistan, which strongly support our current data (Table 3). Faisal et al. [28] reported intra-regional variations in prevalence of *T. gondii* infection, with a rate of 26% and 10% in Gohati and Dagi regions of district Swabi in Khyber Pakhtunkhwa province of Pakistan.. They stated that prevalence of infection was high in women of younger age group. Majid et al. [29] reported that upper Dir had a high prevalence rate 33.03% of chronic Toxoplasma infection as compared to district Lower Dir and Swat in Malakand division of Khyber Pakhtunkhwa, while Sadiqui et al. [33] also reported significant (<0.05) variation in prevalence of Toxoplasma infection in women of Abbottabad and Mansehra district of Hazara division, which supports our current data (Table 2). Khan et al. [37] also stated that there is regional variation in seroprevalence of Toxoplasma infection among pregnant women, with a range of (63%) cases from Punjab province of Pakistan, followed by (48%), (19.25%) and (14.4%) cases from Azad Kashmir, district Sawabi and Kohat district of Khyber Pakhtunkhwa province of Pakistan, which supports our current data (Table 3). Ali [38] stated that environment plays an important role in cross-species transmission and regional prevalence variations of *T. gondii* infection in women in Pakistan. Samudio et al. [6] stated that route of parasite transmission may vary according to human habits in various geographical regions of the world, due to which prevalence of infection may vary in various geographical regions in the world, which supports current variation in prevalence of infection in various districts of Khyber Pakhtunkhwa province of Pakistan (Table 3). The presence of pet cats in homes in the studied regions (Figure 2) could also be not ignored, as these animals are main source for human infection due to close relationship with human communities. A significantly (<0.05) high prevalence of Toxoplasma infection in women of rural areas (Table 1) could be due to high prevalence of infection in cats in rural regions of the province, because infected cats are main reservoir and source of human infection. Ahmad et al. [39] reported Toxoplasma infection with a prevalence of 26.43% in cats brought to different veterinary hospitals and private pet clinics in northern sub-tropical arid region of Pakistan. They also observed that cats in rural areas showed a higher prevalence rate as compared to urban regions. Ahmad and Qayyum, [40] found that presence of cats in vicinity, poor hygienic conditions and extensive animal management systems are responsible for widespread prevalence of *T. gondii* infection in various regions of Pakistan. They further stated that cattle and buffaloes are important sources of milk and meat in the country and risk of infection is high due to lack of modern animal farming system in Pakistan. The results of current observation on regional prevalence variations (Table 3) of *T. gondii* infection are also supported by studies of different international investigators in published literature. Zemene et al. [41], Subauste et al. [42], Flegr et al. [43], Tonouhewa et al. [44] and Maçin et al. [4] also claimed that due to asymptomatic nature of infection, it is difficult to determine exact source of *T. gondii* infection, but hygiene, nutritional habits, geographical region, climate and cultural change cause significant variation in prevalence rate of infection in various countries, geographical regions and even within a particular area in a region.

Assessment of seasonal and annual prevalence of Toxoplasma infection was carried out in women patients (Table 4 and Table 5), but no significant (>0.05) seasonal and annual variation was found in prevalence of infection during three-years studies, which could be due to sampling errors or persistent asymptomatic chronic Toxoplasma infection in women population in the studied regions of Khyber Pakhtunkhwa province of Pakistan (Figure 1). The clinician could not differentiate early, active, latent, chronic, congenital, or reactivated infection from serological assays, such as latex agglutination assay (LAT), immunochromatographic techniques (ICT), and enzyme linked immunosorbant assay (ELISA) (Table 7), which are used for qualitative and quantitative detection of IgM and IgG antibodies, but cannot differentiate between acute and chronic persistent infection of *T. gondii* parasite. Contreras [45] and Thangarajah et al. [46] also support our views and stated that it always remained a diagnostic challenge for clinicians, while Liu et al. [47] and Grzybowski et al. [48], suggested development of specific molecular markers for clinical differentiation of acute stage of *T. gondii* infection from chronic stages. The asymptomatic and chronic nature of *T. gondii* infection poses various diagnostic, therapeutic and preventive challenges. During the current study, latex agglutination test (LAT) and immunochromatographic techniques (ICT) were used as initial screening tests for diagnosis of *T. gondii* infection in women, while enzyme linked immunosorbant assay (ELISA) was used as a confirmatory assay for infection (Table 7), which was based on t qualitative and quantitative detection of (IgM) and (IgG) antibodies in sera of infected patients. A good correlation (<0.05) was found between diagnostic efficacy of LAT and ICT assays, while ELISA significantly (<0.05) confirmed all actual positive cases. Tekkesin [49], Zhang et al. [9], and Hajissa et al. [50] also claimed that due to asymptomatic nature, serological assays are primary, easy and cheap approach for achieving satisfactory outcomes, which also play a crucial role in clinical management of infection, while clinicians usually rely on and accept ELISA results due to its high sensitivities and specificities. These laboratory assays are important for those clinicians, who are working in resource-limited countries like Pakistan and other developing countries.

## 5. Conclusions

The current study of *Toxoplasma gondii* infection in women patients of hospitals, health care, and maternity centers provides important information related to geographical distribution, regional and seasonal variation of infection, and diagnostic efficacy of various serological assays that are used in Khyber Pakhtunkhwa province of Pakistan. The current results revealed that Toxoplasmosis is a widespread infection in women community of the country, while there is no regular screening program, which is a serious issue for pregnant and immune-compromised women in the region. Due to asymptomatic nature of t infection, lack of awareness and national screening program, infection in pregnant women is a potential threat for pregnancy. Moreover, most maternity patients are reluctant to visit physician during pregnancy, which creates public health and socioeconomic problems in women community of endemic region. We recommend that the country should be divided into endemic, sporadic and epidemic regions by modern GIS and remote sensing information technology, which will work like a road map for public and animal health authorities to develop monitoring and controlling strategies for such a neglected zoonotic infection in both human and animals in Pakistan. This will minimize the burden and transmission of disease in Pakistan and across the country.

## Figures and Tables

**Figure 1 tropicalmed-07-00430-f001:**
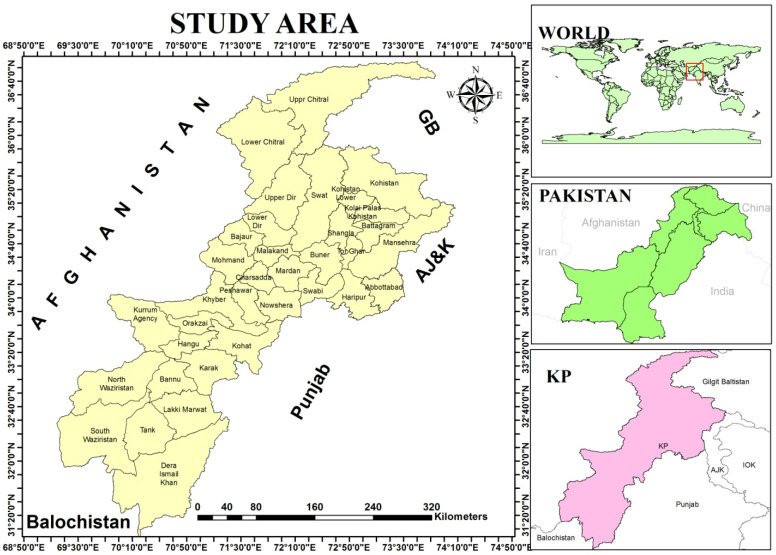
Map of study region, developed by using geographical information system (GIS), which shows various regions (districts) in Khyber Pakhtunkhwa (KP) province of Pakistan.

**Figure 2 tropicalmed-07-00430-f002:**
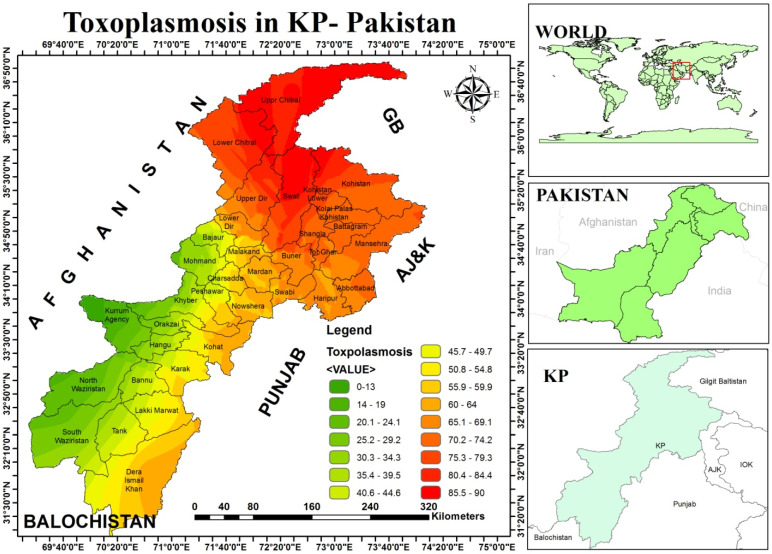
Development of GIS maps, which show the geographical distribution of emerging zoonotic Toxoplasma infection in women patients in various regions of the Khyber Pakhtunkhwa (KP) province of Pakistan.

**Table 1 tropicalmed-07-00430-t001:** Results of Descriptive Socio-demographic Characteristics and Sero-prevalence of *T. gondii* infection in Women Patients in Khyber Pakhtunkhwa Province, Pakistan.

Demographic Characteristics			Prevalence of Infection		95% CI
Host	N	%	Positive	%	
Women	3586		881	24.56	0.2176–0.2744
Maternal Age					*p*-value
17–36 years	3015	84.07	729	24.18	0.09182
37–45 years	571	15.92	152	26.62	----
Residential statues of patients					----
Rural region	2589	72.19	675	26.07	0.0007474
Urban region	997	27.81	206	20.66	----
Obstetric statues of women patients					----
Pregnant	1068	29.78	246	23.03	0.09113
Non-Pregnant	2518	70.22	635	25.21	----

2-sample proportions test.

**Table 2 tropicalmed-07-00430-t002:** Geographical Distributions (Division-wise) of *T. gondii* infection in Women Patients of Khyber Pakhtunkhwa Province, Pakistan.

Divisions in Province	Patients	Prevalence of Toxoplasma Infection		
	N	881	24.56%	*p*-Value
Malakand	1074	287	26.72	1.000
Peshawar	584	80	13.69	1.000
Hazara	609	177	29.06	1.000
Kohat	429	129	30.06	1.000
Mardan	380	81	21.31	1.000
D.I Khan	287	80	27.87	1.000
Bannu	223	47	21.07	1.000
*p*-value			0.000 *	0.000 *

1-sample proportions test. Division vs. Disease type: The Chi-square test for impendence of the two categorical variables (Division and Disease type) was used. * Pearson Chi-Square. * Likelihood Ratio. (* Significant on both statistical test)

**Table 3 tropicalmed-07-00430-t003:** Geographical Distributions (Divisional and District-wise) of *T. gondii* infection in Women Patients of Khyber Pakhtunkhwa, Pakistan.

Division/District	Patients		Prevalence of Infection		
	N	%	Positive	%	*p*-Value
Malakand Division	1074	29.95	287	26.72	1.000
Swat	223	6.22	90	40.36	0.998
Malakand	193	5.38	55	28.49	1.000
Chitral	105	2.93	47	44.76	0.8585
Buner	134	3.74	23	17.16	1.000
Upper Dir	172	4.79	31	18.02	1.000
Lower Dir	140	3.91	24	17.14	1.000
Shangla	107	2.98	17	15.88	1.000
*p*-value	----	----	0.000 *	0.000 **	----
Peshawar Division	584	16.28	80	13.71	1.000
Peshawar	247	6.89	30	12.14	1.000
Charsadda	189	5.27	34	17.98	1.000
Nowshera	148	4.13	16	10.81	1.000
*p*-value	----	----	0.106 *	0.113 **	----
Kohat Division	429	11.97	129	30.07	1.000
Kohat	192	5.35	50	26.04	1.000
Karak	109	3.04	49	44.95	0.854
Hangu	128	3.57	30	23.43	1.000
*p*-value	----	----	0.000 *	0.001 **	----
Hazara Division	609	16.98	177	29.06	1.000
Mansehra	139	3.87	51	36.69	0.9992
Abbottabad	170	4.74	46	27.05	1.000
Battagram	110	3.07	36	32.72	0.9999
Kohistan	85	2.37	26	30.58	0.9998
Haripur	105	2.93	18	17.14	1.000
*p*-value	----	----	0.015 *	0.012 **	----
Mardan Division	380	10.61	81	21.21	1.000
Mardan	239	6.66	52	21.75	1.000
Swabi	141	3.93	29	20.56	1.000
*p*-value	----	----	0.784 *	0.784 **	0.897 ***
D.I Khan Division	287	8.01	80	27.87	1.000
Dera Ismail Khan	188	5.24	64	34.04	1.000
Tank	99	2.76	16	16.15	1.000
*p*-value	----	----	0.001 *	0.001 **	0.001 ***
Bannu Division	223	6.22	47	21.07	1.000
Bannu	120	3.35	33	27.51	1.000
Lakki Marwat	103	2.87	14	13.59	1.000
*p*-value	----	----	0.011 *	0.010 **	0.013 ***

1-sample proportions test. The Chi-square test for impendence of two categorical variables districts and disease type was used. * Pearson Chi-Square. ** Likelihood Ratio. *** Fisher’s Exact Test.

**Table 4 tropicalmed-07-00430-t004:** Prevalence of *T. gondii* infection in Different Months and Seasons in Women Patients of Khyber Pakhtunkhwa Province, Pakistan.

Month and Season			Prevalence of Infection		
**Spring**	**N**	**(%)**	**Positive**	**(%)**	***p*-Value**
	600	16.73	127	21.16	1.000
March	310	8.64	70	22.58	1.000
April	290	8.08	57	19.65	1.000
**Summer**	1501	41.85	380	25.31	1.000
May	373	10.41	109	29.22	1.000
June	251	7.11	54	21.51	1.000
July	329	9.17	40	12.15	1.000
August	307	8.56	97	31.59	1.000
September	241	6.72	80	33.19	1.000
**Autumn** October	232	6.46	58	25.01	1.000
**Winter**	1253	34.94	316	25.21	1.000
November	309	8.62	92	29.77	1.000
December	323	9.01	47	14.55	1.000
January	231	6.44	70	30.31	1.000
February	390	10.87	107	27.43	1.000

1-sample proportions test without continuity correction.

**Table 5 tropicalmed-07-00430-t005:** Prevalence of *T. gondii* infection in Different Seasons in Women Patients in Khyber Pakhtunkhwa, Pakistan.

Variables	Total Patients		Prevalence of Infections		
Various Seasons in Pakistan	N	%	Positive	%	*p*-Value
Spring (March to April)	600	16.73	127	21.16	1.000
Summer (May to September)	1501	41.86	380	25.31	1.000
Autumn (October)	232	6.47	58	25.01	1.000
Winter (November to February)	1253	34.95	316	25.22	1.000
Total	3586		881	24.56	0.211 * 0.201 **

1-sample proportions test. * Pearson Chi-Square (SPSS). ** Likelihood Ratio (SPSS).

**Table 6 tropicalmed-07-00430-t006:** Results of Annual Prevalence of *T. gondii* Infection in Women Patients of Khyber Pakhtunkhwa, Pakistan.

Year	2017			2018				2019		
Month	N	Prevalence %		N	Prevalence %		N	Prevalence %		*p*-Value
March	93	9	9.67	127	43	33.85	90	18	20.00	1.000
April	110	39	35.45	107	11	10.28	73	7	9.58	1.000
May	128	16	12.5	138	62	44.92	107	31	28.97	1.000
June	78	31	39.74	54	10	18.51	119	13	10.92	1.000
July	96	3	3.12	98	7	7.14	135	30	22.22	1.000
August	136	44	32.35	118	47	39.83	53	6	11.32	1.000
September	55	10	18.18	103	29	28.15	83	41	49.39	1.000
October	46	3	6.52	70	18	25.71	116	37	31.89	1.000
November	137	50	36.49	124	31	25.00	48	11	22.91	1.000
December	86	13	15.11	146	9	6.16	91	25	27.47	1.000
January	101	33	32.672	101	23	22.77	29	14	48.27	1.000
February	113	40	35.39	162	37	22.83	115	30	26.08	1.000
Total	1179	291	24.68	1348	327	24.25	1059	263	24.83	0.942 *

3-sample proportions test. * Pearson Chi-Square (SPSS). * Likelihood Ratio (SPSS). * Linear-by-Linear Association (SPSS). (Significant on 3 statistical tests).

**Table 7 tropicalmed-07-00430-t007:** Evaluation of Various Diagnostic assays used for Detection of Toxoplasma infection in Women Patients in Khyber Pakhtunkhwa Province of Pakistan.

Toxoplasma Specific Serological Assays				
Method	N	Prevalence	(%)	*p*-Value
Latex Agglutination Test (LAT)				0.9568 **
LAT	3586	913	25.46	0.000 *
ICT Lateral Flow Chromatography Immunoassay				
ICT	3586	915	25.51	0.000 *
IgM		112	3.12	0.000 ***
IgG		694	19.35	
IgM & IgG		109	3.04	
Enzyme linked immunosorbant assay (ELISA)				
ELISA	1195	881	73.72	0.000 *
IgM		96	8.03	0.000
IgG		682	57.07	
IgM & IgG		103	8.62	

ICT: immunochromatographic test, ELISA: enzyme linked immunosorbant assay. * Pearson Chi-Square. * Continuity Correction ^b^. * Likelihood Ratio. * Fisher’s Exact Test. * Linear-by-Linear Association by SPSS. ** Testing the difference of proportions between LAT and ICT test. ** 2-sample test for equality of proportions without continuity correction. *** 3-sample test for equality of proportions without continuity correction. (* Significant on 4 statistical tests).

## Data Availability

All data generated or analyzed during this work are included in this published article.

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
