# Peer review of "Assessment of Geographical Distribution of Emerging Zoonotic Toxoplasma gondii Infection in Women Patients Using Geographical Information System (GIS) in Various Regions of Khyber Pakhtunkhwa (KP) Province, Pakistan"

_tropicalmed, 2022, doi:10.3390/tropicalmed7120430_

Round 1

Reviewer 1 Report

The study aimed to detect anti-Toxoplasma gondii antibodies in 3586 serum samples of women from Khyber Pakhtunkhwa Province, Pakistan, and assessment the geographical distribution of the infection. It is an important contribution to the field in Pakistan. Suggestions and comments are provided below to help the authors improve the quality and clarity of the paper.

Title suggestion: Assessment of Geographical Distribution of Emerging Zoonotic Toxoplasma gondii Infection in Women Patients Using Geographical Information System (GIS) in Various Regions of Khyber Pakhtunkhwa Province, Pakistan

It is not clear for me that the study analyzed the disease (Toxoplasmosis), that is, the women included in the study had clinical signs compatible with toxoplasmosis and the symptoms were associated with the serological test result. It seems that the study aimed to assess the prevalence of T. gondii infection in women of Khyber Pakhtunkhwa Province, Pakistan. A clinical exam was mentioned (line 183), however, it was not described, and symptoms/disease/test results were not correlated. For this reason, I recommend using the term “Toxoplasma gondii infection”, “Toxoplasma antibodies” or “infection” instead of “Toxoplasmosis” or “disease”. Please, check and replace along the text (e.g. lines 183-189; 194-195, 203-206,…)

Line 25 – T. gondii

Abstract - The abstract should be a single paragraph and should follow the style of structured abstracts, but without headings.

Line 31 – Include dot: Pakistan.

Lines 35-36 – Provide the meaning of ICT and iELISA.

Line 39 – “Among 3586 patients, 881 (24.56%) were found positive.” In which test?

Line 46 – Change to: LAT, ICT and ELISA assays significantly (<0.05) detected Toxoplasma infection.

Line 47 and 215 – Correct: (0.05) – duplicated parenthesis

Line 57 – I recommend choosing keywords that are different from the words in the title.  

Standardize citations formatting according to Instructions for Authors (correct et al.). Suggestion: prevalence,

Line 68 – Replace cats for felids: “…sexual part occurs in domestic and wild felids…”

Introduction – Classic references (as the many papers published by Dubey et al.) could be used in introduction, mainly for well-known information (e.g., lines 64-72), because this information was stated many years ago.

Line 75 – Please, explain the sentence: “Drinking of water contaminated with cysts”. Do you mean oocysts?  Provide the reference for information in lines 75-76. Please, include the consumption of unpasteurized milk as a possibility for transmission.

Line 83-85 – It was used OIE as reference, for this reason: this sentence is related to which species Humans or non-human animals? All paragraph describes the infection in humans, for this reason, I recommend using a properly reference. In addition, I recommend rewriting this sentence because: DAT and MAT are also agglutination tests; histopathology is a complementary test and alone it is not used for confirming the infection.

Line 87 – Replace coma for dot: “region. The”

Line 88 – The term “frequency” could be replaced by “prevalence” whether the authors provide a statistical sample size determination. Replace Toxoplasmosis: Toxoplasma gondii infection in women.

Line 108 and 111 – Khyber Pakhtunkhwa (KP), Pakistan

Figure 2 – I recommend using Toxoplasma infection or Toxoplasma antibodies for the title and legend.

Line 113 – According to the figures, I recommend using Khyber Pakhtunkhwa (KP), or standardize the figures and the legends to KPK. This abbreviation should be used in line 94 (first appearance).

Line 121 – Provide all the abbreviation meanings (LAT, ICT, ELISA).

Line 126 – Why be married was an inclusion criteria in the study?

Line 140 – Mention this is the ICT test.

Line 142 – Please, provide the label/company of the immunochromatography technology-base test used.

Line 144 – “diagnoses of the infection.”

Italicize all the words “Toxoplasma” and “Toxoplasma gondii” in the text (Lines 133, 149, 221, 299…)

Please, cite the precedence of positive and negative controls that were used in the serological tests.

Line 186 – Which test was considered for this result (881; 24.56%)? ELISA?

Lines 190-193 – The authors could use the following format to present the data to assess a better visual analysis of the result: “Among patients from Kohat Division, 30.07% (129/429) cases were found… followed by 29.06% (177/609)… 27.87% (80/287)…”

Lines 212-213 – Please, review the sentence. It seems that information is duplicated.

Line 213 – “…detected Toxoplasma gondii infection.”

Lines 215-216 – How many samples were simultaneously positive by LAT and ICT? All the 913 positive samples by LAT were also positive by ICT? Please, provide the information in the text.

Lines 218-219 – The authors could use the following format to present the data to assess a better visual analysis of the result: “… confirmed 73.72% (881/NN-total of samples analyzed by ELISA) infected women with positivity of IgM (96/ NN-total of samples analyzed by ELISA; 8.03%), IgG (682/ NN-total of samples analyzed by ELISA; 57.07%)….(103/ NN-total of samples analyzed by ELISA; 8.62%”

Line 219 – Please, provide the total number of serum samples analyzed by ELISA.

For all tables (1 to 7) – Replace Toxoplasmosis by “Toxoplasma gondii infection” in the legend and inside the table. Please, mention Table 5 in the text.

Table 1 – The table should be formatted for better comprehension of the data. The division of columns are not clear. Table content (at least the main topics) may be left alignment. In addition, replace “N=3586” by “N”. Include “N” in column/line above 881.

Line 230 – What the authors mean with disease type? Do you mean serological test (ELISA) result: positive or negative?

Line 256 – This is the first time that the authors mention an ICT Immunoblot-based test. Please, provide this information in the properly Materials and Methods section.  

Lines 265-272 – The first paragraph of discussion may be placed in introduction section.

Line 277-278 – frequency of Toxoplasma infection

Line 286 – Replace “chickens used for food”: chickens intended for human consumption…

Line 289 – Remove (it is redundant): “contaminated with oocysts of T. gondii.”

Line 302 – May be the authors would prefer to remove the information “contact with pet dogs as risk factor” because the Reference (Majid et al. 2016) cited “In the studied areas stray cats and dogs were significantly associated with high risk of infection”. In addition, dogs do not play a role in the transmission of T. gondii (at least not directly or significantly).

Line 305-306 – I recommend removing this information “Chaudhary et all., (2006) also stated T. gondii create a significant public health problem in Pakistan due to close contact between humans and various livestock animals.” since close contact of human to livestock animals does not transmit the pathogen and the sentence may be not well understand by the readers.

Line 308 – I understand that the disease was not evaluated, but the prevalence of Toxoplasma gondii infection/antibodies.

Line 315 – status

Line 319-321 – Suggestion: “…earning and they usually drink unpasteurized animal milk, which may…”

Line 327 and 336 – “reported Toxoplasma gondii infection in humans” / “infection prevalence”

Lines 339-340 – Please, rewrite the sentence for better comprehension

Lines 324-360 – The two paragraphs must be summarized and contextualized with the data of the present study. In its current format it is written as a literature review rather than a discussion. As risk factors were not addressed by this study, most of this information could be removed and will not interfere in discussion.

Lines 343-348 – For the same reason as stated for Line 302, I suggest that dogs should not be mentioned in lines 343-348 because they are not an important source of Toxoplasma gondii infection.

Line 375 – based on

Line 384 – 5. Conclusion

Line 385 – The current study of Toxoplasma gondii infection in women…

Line 388 – various serological assays which are used in Khyber Pakhtunkhwa province…                                                            

Author Response

       Para-wise Answer of Reviewer Comments

Reviewer 1 comments

Comments and Suggestions for Authors

The study aimed to detect anti-Toxoplasma gondii antibodies in 3586 serum samples of women from Khyber Pakhtunkhwa Province, Pakistan, and assessment the geographical distribution of the infection. It is an important contribution to the field in Pakistan. Suggestions and comments are provided below to help the authors improve the quality and clarity of the paper.

Title suggestion: Assessment of Geographical Distribution of Emerging Zoonotic Toxoplasma gondii Infection in Women Patients Using Geographical Information System (GIS) in Various Regions of Khyber Pakhtunkhwa Province, Pakistan. It is not clear for me that the study analyzed the disease (Toxoplasmosis), that is, the women included in the study had clinical signs compatible with toxoplasmosis and the symptoms were associated with the serological test result. It seems that the study aimed to assess the prevalence of T. gondii infection in women of Khyber Pakhtunkhwa Province, Pakistan. A clinical exam was mentioned (line 183), however, it was not described, and symptoms/disease/test results were not correlated. For this reason, I recommend using the term “Toxoplasma gondii infection”, “Toxoplasma antibodies” or “infection” instead of “Toxoplasmosis” or “disease”. Please, check and replace along the text (e.g. lines 183-189; 194-195, 203-206,…)

Answer:As per reviewer-1 recommendation, The term Toxoplasmosis was replaced by world Toxoplasma gondii infection in the title of article as well as in lines 183-189; 194-195, 203-206 and other lines of the article.

Q: 2 Line 25 – T. gondii

Answer: Line 25 was corrected by using T. gondii word instead of Toxoplasmosis

Q: Abstract - The abstract should be a single paragraph and should follow the style of structured abstracts, but without headings.

Answer: The abstract was modified and corrected

Q: Line 31 – Include dot: Pakistan.

Answer: Line 31 was corrected as per suggestion

Q: Lines 35-36 – Provide the meaning of ICT and iELISA.

Answer: Meaning of ICT and ELISA was included in the article.

Q: Line 39 – “Among 3586 patients, 881 (24.56%) were found positive.” In which test?

Answer: It was corrected. Overall prevalence of infection was 881 (24.56%). Among 3586 women, true positive cases were 881(24.56%).

Q: Line 46 – Change to: LAT, ICT and ELISA assays significantly (<0.05) detected Toxoplasma infection.

Answer: It was changed as per recommendation

Q: Line 47 and 215 – Correct: (0.05) – duplicated parenthesis

Answer: Duplicated parenthesis was removed from line 47 and 215

Q: Line 57 – I recommend choosing keywords that are different from the words in the title. 

Answer: Keywords were changed from Geographical distribution, Toxoplasmosis, Women patients, Khyber Paktonkhwas, province Pakistan to Zoonotic infection, Diagnosis, Regional variations, Divisions and Districts

Q: Standardize citations formatting according to Instructions for Authors (correct et al.). Suggestion: prevalence,

Answer: Citations were changed from et all., to et al. as per reviewer recommendation.

Q: Line 68 – Replace cats for felids: “…sexual part occurs in domestic and wild felids…”

Answer: Line 68 was corrected as per reviewer suggested words.

Q: Introduction – Classic references (as the many papers published by Dubey et al.) could be used in introduction, mainly for well-known information (e.g., lines 64-72), because this information was stated many years ago.

Answer: More sentences were added in introduction portion of the article.

Q: Line 75 – Please, explain the sentence: “Drinking of water contaminated with cysts”. Do you mean oocysts?  Provide the reference for information in lines 75-76. Please, include the consumption of unpasteurized milk as a possibility for transmission.

Answer: The above mentioned sentence was included in introduction part of the article

Q: Line 83-85 – It was used OIE as reference, for this reason: this sentence is related to which species Humans or non-human animals? All paragraph describes the infection in humans, for this reason, I recommend using a properly reference. In addition, I recommend rewriting this sentence because: DAT and MAT are also agglutination tests; histopathology is a complementary test and alone it is not used for confirming the infection.

Answer: Line 83-85 was corrected in introduction portion of the article

Q: Line 87 – Replace coma for dot: “region. The”

Answer: Comma was replace for dot in line 87

Q: Line 88 – The term “frequency” could be replaced by “prevalence” whether the authors provide a statistical sample size determination. Replace Toxoplasmosis: Toxoplasma gondii infection in women.

Answer: The term “frequency was replaced in line 88 and other part of the article

Q: Line 108 and 111 – Khyber Pakhtunkhwa (KP), Pakistan

Answer: Khyber Pakhtunkhwa (KP), Pakistan was added in line 108 and 111 and in other paragraphs of the article

Q: Figure 2 – I recommend using Toxoplasma infection or Toxoplasma antibodies for the title and legend.

Answer: Figure 2 was corrected with above words.

Q: Line 113 – According to the figures, I recommend using Khyber Pakhtunkhwa (KP), or standardize the figures and the legends to KPK. This abbreviation should be used in line 94 (first appearance).

Answer: Khyber Pakhtunkhwa (KP) was added in the figures and the legends to KPK. This abbreviation was also used in line 94 (first appearance).

Q: Line 121 – Provide all the abbreviation meanings (LAT, ICT, ELISA).

Answer: Meanings of all abbreviation (LAT, ICT, ELISA) was mentioned in line 121 and in whole article

Q: Line 126 – Why be married was an inclusion criteria in the study?

Answer: The study was carried in married women, who visited hospital for reproductive problems, infertility or adverse pregnancy outcomes.

Q: Line 140 – Mention this is the ICT test.

Answer: Line 140 was explained

Q: Line 142 – Please, provide the label/company of the immunochromatography technology-base test used.

Answer: Fortress Diagnostics Limited, UK

Q: Line 144 – “diagnoses of the infection.”

Answer: Line 144 was corrected by adding above mentioned words.

Q: Italicize all the words “Toxoplasma” and “Toxoplasma gondii” in the text (Lines 133, 149, 221, 299…)

Answer: All the words of Toxoplasma gondii have Italicized including (Lines 133, 149, 221, 299…)

Q: Please, cite the precedence of positive and negative controls that were used in the serological tests.

Answer: Positive and negative control was corrected 

Q: Line 186 – Which test was considered for this result (881; 24.56%)? ELISA?

Answer: (881;24.56%) was true positive cases in 3586 women and overall prevalence of infection in 3586 women, which was more explained in result portion of article.

Q: Lines 190-193 – The authors could use the following format to present the data to assess a better visual analysis of the result: “Among patients from Kohat Division, 30.07% (129/429) cases were found… followed by 29.06% (177/609)… 27.87% (80/287)…”

Answer: The result portion of article was completely converted into format suggested by reviewer.

Q: Lines 212-213 – Please, review the sentence. It seems that information is duplicated.

Answer: Line 212-213 was reviewed and changed.

Q: Line 213 – “…detected Toxoplasma gondii infection.”

Answer:  Line 213 was corrected by adding the above mentioned words.

Q: Lines 215-216 – How many samples were simultaneously positive by LAT and ICT? All the 913 positive samples by LAT were also positive by ICT? Please, provide the information in the text.

Answer: These information have been added into last paragraphs of result portion of the article.

Q: Lines 218-219 – The authors could use the following format to present the data to assess a better visual analysis of the result: “… confirmed 73.72% (881/NN-total of samples analyzed by ELISA) infected women with positivity of IgM (96/ NN-total of samples analyzed by ELISA; 8.03%), IgG (682/ NN-total of samples analyzed by ELISA; 57.07%)….(103/ NN-total of samples analyzed by ELISA; 8.62%”

Answer: The result portion of article have been converted into above mentioned format recommended by reviewer

Q: Line 219 – Please, provide the total number of serum samples analyzed by ELISA.

Answer: The information have been added into Table No:7

Q: For all tables (1 to 7) – Replace Toxoplasmosis by “Toxoplasma gondii infection” in the legend and inside the table. Please, mention Table 5 in the text.

Answer: For all tables (1 to 7) Toxoplasmosis was replaced by “Toxoplasma gondii infection” in the legend and inside the table. Table 5 was also mentioned in the text of result and discussion portion of the article.

Q: Table 1 – The table should be formatted for better comprehension of the data. The division of columns are not clear. Table content (at least the main topics) may be left alignment. In addition, replace “N=3586” by “N”. Include “N” in column/line above 881.

Answer: Table:1 was corrected

Q: Line 230 – What the authors mean with disease type? Do you mean serological test (ELISA) result: positive or negative?

Answer: Line 230 was corrected

Q: Line 256 – This is the first time that the authors mention an ICT Immunoblot-based test. Please, provide this information in the properly Materials and Methods section.

Answer: corrected  

Q: Lines 265-272 – The first paragraph of discussion may be placed in introduction section.

Answer:  Lines 265-272 – The first paragraph of discussion has been placed in introduction section.

Q: Line 277-278 – frequency of Toxoplasma infection

Answer: Line 277-278 – frequency of Toxoplasma infection has converted into word prevalence in line 277-278

Q: Line 286 – Replace “chickens used for food”: chickens intended for human consumption…

Answer: Line 286 – line “chickens used for food”: has changed into chickens intended for human consumption…

Q: Line 289 – Remove (it is redundant): “contaminated with oocysts of T. gondii.”

Answer: Line 289 – The word has corrected “contaminated with oocysts of T. gondii

Q: Line 302 – May be the authors would prefer to remove the information “contact with pet dogs as risk factor” because the Reference (Majid et al. 2016) cited “In the studied areas stray cats and dogs were significantly associated with high risk of infection”. In addition, dogs do not play a role in the transmission of T. gondii (at least not directly or significantly).

Answer: The irrelevant words/ information related to transmission of T. gondii has been removed

Q:  Line 305-306 – I recommend removing this information “Chaudhary et all., (2006) also stated T. gondii create a significant public health problem in Pakistan due to close contact between humans and various livestock animals.” since close contact of human to livestock animals does not transmit the pathogen and the sentence may be not well understand by the readers.

Answer: The above information has been removed from line 305-306

Q: Line 308 – I understand that the disease was not evaluated, but the prevalence of Toxoplasma gondii infection/antibodies.

Answer: Yes prevalence of Toxoplasma gondii infection was found

Q: Line 308

Answer: Line 308 has corrected

Q: Line 315 – status

Answer: Line 315 has corrected

Q: Line 319-321 – Suggestion: “…earning and they usually drink unpasteurized animal milk, which may…”

Answer: suggested words has included in line 319-321

Q: Line 327 and 336 – “reported Toxoplasma gondii infection in humans” / “infection prevalence”

Answer: Line 327 and 336 has corrected with above mentioned words

Q: Lines 339-340 – Please, rewrite the sentence for better comprehension

Answer: The sentences has been rewritten in line 339-340

Q: Lines 324-360 – The two paragraphs must be summarized and contextualized with the data of the present study. In its current format it is written as a literature review rather than a discussion. As risk factors were not addressed by this study, most of this information could be removed and will not interfere in discussion.

Answer: Lines 324-360 has been changed summarized, while some words has been removed from paragraphs.

Q: Lines 343-348 – For the same reason as stated for Line 302, I suggest that dogs should not be mentioned in lines 343-348 because they are not an important source of Toxoplasma gondii infection.

Answer: Lines 343-348 has corrected as per recommendation of reviewer

Q: Line 375 – based on

Answer: Line 375 has corrected

Q: Line 384 – 5. Conclusion

Answer: line 384 has Corrected

Q: Line 385 – The current study of Toxoplasma gondii infection in women

Answer: Line 385 has been corrected

Q: Line 388 – various serological assays which are used in Khyber Pakhtunkhwa province…                       Answer: Line 388 has been corrected  

Reviewer 2 Report

This study collected 3586 serum samples from women in Pakistan. They used three methods to detect T. gondii antibodies in the sera and found that 881(24.56%) of the patients are infected with T. gondii. 

The study provided a wealth of data but failed to give the data due attention and analysis. The data was presented in the tables but was hardly mentioned in the results and examined in the discussion. 

The results session can be significantly improved by presenting the data shown in the tables in the texts in greater detail. It feels like it was rushed. 

The discussion can be significantly revised. Your discussion is taken over by literature review rather than data analysis and extrapolation. Yes. it's useful to talk about previous publications but how does your data present new insights? 

Author Response

Reviewer 2 comments

 Comments and Suggestions for Authors

Q: This study collected 3586 serum samples from women in Pakistan. They used three methods to detect T. gondii antibodies in the sera and found that 881(24.56%) of the patients are infected with T. gondii. 

Answer: The result portion of article has re-written and explained in detail along with prevalence of infection by all three methods used in article. The actual and overall  prevalence of infection is 881(24.56%) that is (881/3586)

Q:The study provided a wealth of data but failed to give the data due attention and analysis. The data was presented in the tables but was hardly mentioned in the results and examined in the discussion. 

Answer: As per suggestion, the result portion of article was rewritten and explained in great details. All table has completely explained in result portion. All tables have discussed in discussion portion of the article

Q: The results session can be significantly improved by presenting the data shown in the tables in the texts in greater detail. It feels like it was rushed.

Answer: As per suggestion, the result portion has significantly improved and explained in detail in text. The result portion has completely re-written and each table was seperatly explained in text of the result portion. 

Q: The discussion can be significantly revised. Your discussion is taken over by literature review rather than data analysis and extrapolation. Yes. it's useful to talk about previous publications but how does your data present new insights? 

Answer: As per recommendation, the discussion portion has significantly revised and changed, All table in result portion of the article has step-wise discussed in discussion portion of the article. The current data in table has been para-wise analyzed and discussed with previous research articles. 

Reviewer 3 Report

The article has serious flaws. The work did not show significant differences between evaluated parameters, as rural or urban area, age and frequency in different months or seasons. Only differences in geographical distributions were observed, however the data did not support one hipothesis for this difference. Besides that, the advantages and disadvantages of diagnosis tests should be approached.

Author Response

Reviewer 3 comments

Comments and Suggestions for Authors

Q: The article has serious flaws. The work did not show significant differences between evaluated parameters, as rural or urban area, age and frequency in different months or seasons. Only differences in geographical distributions were observed, however the data did not support one hypothesis for this difference. Besides that, the advantages and disadvantages of diagnosis tests should be approached.

Answer: The abstracts and results have been rewritten in detail and explained each Table separately. The current study map various districts and division in the whole province of Khyber Pakhtunkhwa, Pakistan, which is a great contribution and will help in control/ eradiation of infection in human and animal. The Table of various diagnostic tests (Table:7) has been rearranged and explained in detail in the text in result portion and discussion portion of the article

Round 2

Reviewer 2 Report

I still don't understand the meaning of the column(%) followed by the N(total number). It should be clearly stated in the table description below what the denominator is for the percentages. 

I have trouble understanding table 6. The first column is N(%). What does that mean? What are the numerator and denominator? 

Author Response

Reviewer 2

Para-wise correction of Reviewer 2

Comments and Suggestions for Authors

Q: I still don't understand the meaning of the column (%) followed by the N (total number). It should be clearly stated in the table description below what the denominator is for the percentages?

Answer: The Table has corrected and made clear for readers. The World “N” means total samples collected per month in the studied year. The symbol (%) has removed from column containing (N) and has mentioned with column of prevalence.

Q; I have trouble understanding table 6. The first column is N (%). What does that mean? What are the numerator and denominator?

Answer: The Table No 6 has modified and corrected and made clear for easily understanding. The numerators are total patients examined and screen out per month, while patient positive for Toxoplasmosis infection in each month is the denominator.

Reviewer 3 Report

The authors improved the discussion of the data analysis and provided an association with T. gondii soroprevalence. Thus, this manuscript is suitable for publication in TropicalMed.

Author Response

Answer to reviewer 3 comments

Sir thank you so much for your kind approval for acceptance of our manuscript. It's so nice of you.

Kind regards 
